# Brugada Syndrome: Oligogenic or Mendelian Disease?

**DOI:** 10.3390/ijms21051687

**Published:** 2020-03-01

**Authors:** Michelle M. Monasky, Emanuele Micaglio, Giuseppe Ciconte, Carlo Pappone

**Affiliations:** Arrhythmology Department, IRCCS Policlinico San Donato, Piazza E. Malan 1, San Donato Milanese, 20097 Milan, Italy; michelle.monasky@grupposandonato.it (M.M.M.); emanuele.micaglio@grupposandonato.it (E.M.); g.ciconte@gmail.com (G.C.)

**Keywords:** Brugada syndrome, sudden cardiac death, genetic testing, mutation, *SCN5A*, sodium channel, arrhythmia, channelopathy, segregation analysis, functional studies

## Abstract

Brugada syndrome (BrS) is diagnosed by a coved-type ST-segment elevation in the right precordial leads on the electrocardiogram (ECG), and it is associated with an increased risk of sudden cardiac death (SCD) compared to the general population. Although BrS is considered a genetic disease, its molecular mechanism remains elusive in about 70–85% of clinically-confirmed cases. Variants occurring in at least 26 different genes have been previously considered causative, although the causative effect of all but the *SCN5A* gene has been recently challenged, due to the lack of systematic, evidence-based evaluations, such as a variant’s frequency among the general population, family segregation analyses, and functional studies. Also, variants within a particular gene can be associated with an array of different phenotypes, even within the same family, preventing a clear genotype–phenotype correlation. Moreover, an emerging concept is that a single mutation may not be enough to cause the BrS phenotype, due to the increasing number of common variants now thought to be clinically relevant. Thus, not only the complete list of genes causative of the BrS phenotype remains to be determined, but also the interplay between rare and common multiple variants. This is particularly true for some common polymorphisms whose roles have been recently re-evaluated by outstanding works, including considering for the first time ever a polygenic risk score derived from the heterozygous state for both common and rare variants. The more common a certain variant is, the less impact this variant might have on heart function. We are aware that further studies are warranted to validate a polygenic risk score, because there is no mutated gene that connects all, or even a majority, of BrS cases. For the same reason, it is currently impossible to create animal and cell line genetic models that represent all BrS cases, which would enable the expansion of studies of this syndrome. Thus, the best model at this point is the human patient population. Further studies should first aim to uncover genetic variants within individuals, as well as to collect family segregation data to identify potential genetic causes of BrS.

## 1. Introduction

Brugada syndrome (BrS) is a cardiac arrhythmia associated with an increased risk of sudden cardiac death (SCD) compared to the general population, diagnosed by the presence of a type 1 BrS pattern on the electrocardiogram (ECG), namely a coved-type ST-segment elevation in the right precordial leads [1,2]. This type 1 pattern may occur spontaneously, intermittently, and in a dynamic way, often in relation with many environmental factors, including drugs, illicit drugs, alcohol, fever, and heavy meals.

In patients in whom a spontaneous type 1 pattern has not been observed, a pharmacological challenge may be performed to unmask the pattern. However, it is imperative that these procedures are performed in specialized centers, due to the associated risks [3]. Current guidelines state that, for BrS, implantable cardioverter defibrillator (ICD) placement is the only proven effective therapy for the prevention of SCD [4], although recent studies have highlighted the potential of radiofrequency ablation of the arrhythmogenic substrate [5].

Although the prevalence has been described as 1:2000 in Western Europe and the USA and 1:500 in Southeast Asia [6], the true prevalence of BrS is unknown, due to the lack of symptoms for even decades in many people and the challenges involved in diagnosis. In Southeast Asia, BrS seems to affect almost exclusively male adults [7]. Pooled analyses indicated that a spontaneous type 1 ECG is an independent risk factor for SCD in males, but not in females, and that male patients are at higher risk of arrhythmic events [8,9]. However, BrS is found throughout the world and in both genders, to whom studies have shown it is transmitted equally [5] as an autosomal dominant disease with incomplete penetrance [10,11,12]. However, a few recent articles suggest some possible alternative mechanisms of inheritance, such as autosomal recessive [13] or X linked [14]. Due to the increasing number of variants, both common and rare, found in BrS patients, perhaps an oligogenic model should replace the traditional Mendelian model for BrS [15].

## 2. The Challenges Surrounding BrS Genetics

To the best of current knowledge, BrS is still considered a genetic disease, although the genetics remain elusive in about 70–85% of clinically-confirmed cases, in spite of the widespread use of the next-generation sequencing (NGS) technique with a high coverage (at least 100×). Variants occurring in at least 26 different genes (Table 1) have been previously considered causative [16], although the causative effect of all but the *SCN5A* gene has been recently challenged [17]. In fact, many genetic tests reporting variants in these genes return results with uncertain significance. The source of such uncertainty is often the aforementioned idea that all BrS cases are inherited with a Mendelian autosomal dominant mechanism. This idea prevents the geneticist from considering a possible cumulative role of both common and rare genetic variants, because, according to the previous hypothesis, there must be one and only one mutation. Thus, the role of cumulative genetic variants within the same individual in the causative effect of disease expression is currently a source of debate [18]. Further complicating matters, different variants within a particular gene can be responsible for an array of different phenotypes [19], even within the same family [20,21]. These situations make many genotype-phenotype correlations very difficult. 

A study by Risgaard and colleagues [22] evaluated the presence in the general population of previously BrS-associated heterozygous variants in several genes, namely *CACNA1C*, *CACNA2D1*, *CACNB2*, *GPD1L*, *KCND3*, *KCNH2*, *SCN1Bb*, *SCN3B*, *KCNJ8*, and *SCN5A*. The authors concluded that variants in these genes are common in the general population, occurring at a rate of 1:23. This article reflects the idea of BrS as a Mendalian disease and used in silico predictions to define the pathogenicity of certain variants. However, in silico predictions cannot be relied upon as a single tool to predict pathogenicity [23]. Nevertheless, this study laid the groundwork in understanding the prevalence of these variants in the general population, which could be used by later groups to develop polygenic risk scores, taking into account other studies that suggest that BrS is very likely a multigenic disorder, rather than a Mendelian condition. Another study by the same group [24] evaluating the clinical picture in patients harboring previously reported BrS-associated genetic variants in the genes *CACNA1C*, *CACNA2D1*, *CACNB2*, *KCNH2*, *PKP2*, *SCN10A*, *SCN1B*, *SCN3B*, *SCN5A*, and *TRPM4* found that the mean J-point elevation in V1 and V2 were within normal limits, and there was no difference in reported incidences of syncope, ventricular arrhythmias, or overall mortality, compared to non-carriers of the variants, concluding that the variants are not the monogenic cause of BrS. However, in that study, patients were not tested for BrS with a provocative drug, and so the spontaneous J-point elevations reported may be misleading. In fact, in that study, no significant differences in J-point elevation were found even between carriers and non-carriers of *SCN5A* variants. Then, another genome-wide association study published by the same group the next year [25] reported an association between the single nucleotide polymorphism (SNP) rs6800541 in the *SCN10A* gene with an increase in J-point elevation compared to wildtype in both lead V1 and V2, while the SNPs studied in the genes *SCN5A* and *HEY2* did not significantly affect the J-point. The SNPs in all three genes *HEY2*, *SCN5A*, and *SCN10A* were associated with significant changes in PR interval and QRS duration. The *SCN5A* and *SCN10A* variants studied were different from those in the prior [24] study. The rs9388451 genetic locus adjacent to the *HEY2* gene was also associated with ventricular fibrillation and cardiac arrest [25]. It is interesting that, again, none of the SNPs studied, including that in the *SCN5A* gene, were found to be predisposing to syncope, atrial fibrillation, or total mortality. However, again, the electrocardiographic data may be misleading, as in the prior study, because it relies on spontaneously collected electrocardiograms, which are well known to be unreliable in the diagnosis of BrS, even for about 80% of patients who have experienced cardiac arrest or syncope because of documented ventricular fibrillation [5,26]. A study by a different group [27] studying the same genetic variation (rs9388451) adjacent to the *HEY2* gene reported its role in the alteration of ion channel expression across the cardiac ventricular wall and its possible association with BrS. Thus, further understanding of the various individual variants within the diverse alleged BrS-related genes is necessary [28].

The search for other BrS-related genes is complicated by several factors. First, as described above, BrS appears to not be a Mendelian disease, but rather an oligogenic disease, which is affected by several loci, each of which is influenced by a huge number of environmental factors. Second, entire genome analysis is extremely costly, and even if it is performed, data from several patients need to be pulled to begin to understand which variants may be causative of BrS versus which variants may be modulators of the syndrome or not related at all to BrS. 

To the best of our knowledge, variants in at least 26 different genes have been previously indicated as causative of BrS, but many genetic factors, such as polymorphisms [29], and non-genetic factors, such as fever, are known to alter the disease expression. Likely, BrS should no longer be described as a single disorder, but rather defined as a group of disorders clumped together only by their common alteration in the ECG [2,30], all characterized by vastly different clinical pictures and patterns of inheritance [20]. 

BrS can share some genetic mutations with other forms of arrhythmias, and this is much more common than had been expected [31]. The same variant can apparently result in different phenotypes even among family members [15,32,33]. And finally, a further source of complexity in discovering other BrS-related genes is that the same BrS phenotype can be caused by variants even in different genes encoding proteins with completely different functions. Variants in some of those genes can even result in overlap syndromes, and, in fact, overlap between BrS and other heart diseases, such as hypertrophic cardiomyopathy [34] and arrhythmogenic right ventricular (RV) dysplasia/cardiomyopathy (ARVD/C) [35], has been described. The limit of genotype-phenotype correlation among these mutations is the incomplete knowledge about the real mechanism of BrS. Today, for instance, it is well understood that the so called “trigger situations” like alcohol, fever, high temperature, drugs and illicit drugs can modulate sodium channel function in the heart conduction system [36], eliciting the BrS ECG pattern even in asymptomatic individuals harboring heterozygous variants and/or mutations in the *SCN5A* gene. An excellent article suggested that, in BrS patients, a certain kind of “metabolic impairment” might exist, and this impairment affects alcohol metabolism [37]. On the other hand, when studying modifiers of the *SCN5A* gene, it is poorly understood how a mutation in a structural gene, encoding for instance a desmosomal protein, can affect channel function enough to cause BrS in the absence of a mutation or variant in the *SCN5A* gene. Variants in genes encoding for desmosomal proteins, such as *JUP*, *DSP*, *PKP2*, and *DSG2*, have been associated with ARVD/C and dilated cardiomyopathy (DCM), both of which are associated with SCD [38]. Desmosomal proteins, such as plakophilin-2 (encoded by *PKP2*) and desmoglein-2 (encoded by *DSG2*), have been described to interact with Na_V_1.5 [39] and implicated as a possible cause of BrS and ARVD/C [40]. However, the role of these desmosomal proteins in BrS pathology is still under debate. 

Since SCD can be the first manifestation of BrS, it is imperative to identify even completely asymptomatic patients with this syndrome, who may not realize the need to visit a physician. It would be extremely useful to have a more powerful genetic test, which could be used on a wide basis, to diagnose the syndrome without drug challenge tests and to stratify the risk of future arrhythmic events [41].

## 3. Sodium Channel Mutations

The most commonly mutated gene in BrS is *SCN5A*, occurring in about 15–30% of cases [72]. Consequently, the Na_V_1.5 encoded protein is the best known in BrS studies. A multi-study analysis consisting of 1892 BrS patients concluded that symptomatic patients at the time of diagnosis or electrophysiological study (EPS) with *SCN5A* variants were at higher risk of arrhythmic events compared to symptomatic *SCN5A*-negative patients [73]. Also, some articles have described the possible modulation of *SCN5A* mutations by common polymorphisms [74]. A common genetic factor that unifies all *SCN5A*-related cases has yet to be found, possibly because BrS is only a particular ECG pattern caused by multiple factors [75] and not by a single mutation. In fact, the normal QRS complex in a human ECG is regulated by several genes, some of them already studied as possible candidates for BrS and, more generally, for familial arrhythmic disorders [76]. Interestingly, this article [76] proposed *SCN10A* as the major QRS regulatory gene and, about eight years later, our research group demonstrated that a comparison between *SCN5A* and *SCN10A* Brugada patients is possible [77]. 

The idea that BrS is only a particular ECG pattern caused by multiple factors and not by a single mutation is supported by the finding that sodium channel blockers do not provoke the BrS ECG pattern in healthy individuals or in all *SCN5A* mutation carriers [75]. Several studies have reported a role for common variants in BrS, particularly in sodium channel-related genes. Genome-wide association studies are particularly important for identifying common modifier variants that alter disease susceptibility. Along these lines, a genome-wide association study identified an association with BrS for two alleles on chromosome 3 and one allele on chromosome 6, located closest to the genes *SCN5A*, *SCN10A*, and *HEY2*, respectively [18]. This demonstrates that predisposition to BrS can occur because of the presence of common genetic variants in sodium channel genes or the transcriptional regulator *HEY2*. Furthermore, association data from genome-wide association studies used to calculate weighted polygenic risk scores have been explored as predictive tools to anticipate the results of ajmaline challenges, with a study by Tadros and colleagues [78] describing the association between polygenic risk scores and the slowing of cardiac conduction with ajmaline. Thus, the role of common variants in disease susceptibility must be considered, and is much more important than previously recognized. 

Routinely, in silico predictions are used to ascertain the likelihood of the pathogenicity of a particular variant. However, these analyses tools are often unreliable or result in an uncertain significance of a particular variant [77]. Thus, family segregation analysis and functional studies are still necessary to understand the likelihood of pathogenicity of a particular variant, even after the performance of in silico studies. 

Given the uncertain significance of many variants found in BrS patients, including the variability between people with the same variant and the lack of functional studies in most cases, recent studies have focused on understanding the phenotypic effect of individual variants within the *SCN5A* gene [16,19,28,42,43,77,79,80], as well as the search for additional genes involved in this multi-causative pathology [32,77,81,82,83]. As previously mentioned, one such study demonstrated the similarity in phenotype between patients harboring *SCN10A* variants, as opposed to *SCN5A* variants, including personal history of cardiac arrest/syncope, spontaneous BrS electrocardiogram pattern, family history of sudden death, and arrhythmic substrate [77]. This is consistent with functional studies performed in human-induced pluripotent stem cell-derived cardiomyocytes, in which single-cell phenotype features of BrS were seen in cells from a patient harboring the variants NM_006514.3:c.3803G>A and NM_006514.3:c.3749G>A in the *SCN10A* gene [47]. In a multicenter study in which candidate genes were sequenced, including *SCN10A*, the study concluded that the common rs6795970 in the *SCN10A* gene was strongly associated with BrS and resulted in a loss of function of Na_V_1.8, as did rare *SCN10A* variants found in patients, although co-segregation studies did not always support the functional study findings [45]. Thus, the study concluded that their data do not support a strong role for *SCN10A* variants as monogenic causes of BrS. However, a study by Hu and colleagues [46] identified *SCN10A* as a major susceptibility gene for BrS, identifying *SCN10A* mutations in 25 out of 150 probands (17%), suggesting an important role for this gene in BrS. The importance of this gene is supported by studies about the influence of the *SCN10A* gene in both cardiac conduction [84] and the autonomic nervous system [85].

The sodium channel genes *SCN1B*, *SCN2B*, and *SCN3B* have been associated with BrS and are included in BrS diagnostic panels, although their role is disputed [17]. A study by Watanabe and colleagues has described the *SCN1B* gene in association with both cardiac conduction disease and Brugada syndrome [86]. Described in case reports [87,88], the R214Q variation in *SCN1Bb* has been reported as a functional polymorphism that may serve as a modifier of the substrate responsible for BrS via a combined loss of function of sodium channel current and gain of function of transient outward potassium current [87]. In another study, the IVS3+ 2996(TTA)8 allele was described as an *SCN1B* polymorphism that may make middle-aged, male Japanese more susceptible to BrS, while not causing BrS by itself [89]. Furthermore, a study by Yuan and colleagues [49] identified the H162P mutation in the *SCN1Bb* gene in a BrS patient, and, extrapolating that mutation to in vitro studies, found that this mutation reduces the action potential amplitude and conduction velocity, creating an increased risk of ventricular arrhythmia. However, genotype–phenotype correlations in families with BrS and reported pathogenic *SCN1B* or *SCN1Bb* variants are lacking [48]. Additionally, studies report identifying variants in the *SCN1B* gene in BrS patients with low prevalence [90] and low evidence of pathogenicity [17].

Described in a case report, the *SCN2B* gene was identified as a new candidate gene for BrS, reducing sodium channel current density and Na_V_1.5 cell surface expression [51]. *SCN2B* deletion in mice results in ventricular and atrial arrhythmias [91]. Likewise, an *SCN3B* variant described by Hu and colleagues in a case study was suggested to lead to a loss of transport and functional expression of the hNa_V_1.5 protein, resulting in a BrS phenotype [92]. Similarly, a report by Ishikawa and colleagues identified the V110I variant in the *SCN3B* gene in three out of 178 Japanese BrS patients, and demonstrated that this variant leads to decreased cell surface expression of Na_V_1.5 and reduced sodium current [52]. However, segregation analysis is lacking for each of these genes, and further studies should be performed to further understand their effects [17]. 

Sodium channel function can be affected by factors outside of genetic mutations in the gene coding for the channel itself. In one study investigating the consequences of K_V_4.3 overexpression on Na_V_1.5 current and consequent sodium channel availability, the authors concluded that the current of the Na_V_1.5 protein was directly impacted by several factors, including the gain-of-function of the K_V_4.3 protein encoded for by the *KCND3* gene [62]. It has been suggested that post-translational modifications, such as a defect in the splicing process [93] or trafficking [94,95], or a modification in phosphorylation, methylation, or acetylation [96], could explain alterations in the function of the channel encoded by *SCN5A* in the absence of mutations in this gene itself. Also, the predicted phenotypic effect of a particular variant should take into consideration that ancestry can affect the pathogenicity of a particular variant [97]. Thus, there are several factors that can contribute to channel function, or dysfunction, other than mutations in the *SCN5A* gene itself that encodes for the Na_V_1.5 protein.

Another gene currently in the BrS diagnostic panel is *RANGRF* [98], the RAN Guanine Nucleotide Release Factor, also called MOG1, which regulates the expression and function of the Na_V_1.5 cardiac sodium channel in humans by enhancing the expression of Na_V_1.5 at the cell membrane, increasing sodium current densities [53,54,55]. The *GPD1L* gene, which encodes glycerol-3-phosphate dehydrogenase-1 like protein and is currently also included in BrS diagnostic panels [98], also regulates Na_V_1.5. Variants in *GPD1L* have been described as responsible for a loss of enzymatic function resulting in glycerol-3-phosphate PKC-dependent phosphorylation of *SCN5A* at serine 1503, prominently decreasing sodium current [56]. NAD^+^ has been reported to possibly counteract the effect of PKC by activating PKA [99]. However, the causative effect of variants in each of these genes has also been disputed [17].

Some centers now consider the *LRRC10* gene to be associated with BrS [100]. This gene encodes for the Heart-Restricted Leucine-Rich Repeat Protein and has been described as a transcriptional target of Nkx2.5, which regulates the ion channel proteins encoded by the *SCN5A*, *CACNA1C*, and *KCNH2* genes [101].

These studies show that the function of the Na_V_1.5 protein may be affected in diverse ways, either by direct variants in the *SCN5A* gene encoding for Na_V_1.5, or by variants in genes encoding for proteins that interact with Na_V_1.5, modulating its function in a transient way. However, further studies are required to clarify the role of variants in each of these genes in the expression of the BrS phenotype.

## 4. Calcium Channel Mutations

Given the important role of calcium for the cardiac action potential, it is very likely that the real role of calcium currents in BrS is underrated [2]. Calcium is central to excitation-contraction coupling, linking the electrical signal detected by ECG that defines BrS to the mechanical dysfunction, including ventricular fibrillation and reduced contractility, seen in BrS. Importantly, the BrS phenotype is modulated by non-genetic factors, such as an increase in vagal tone or body temperature, factors that are known to display altered calcium signaling. Furthermore, the BrS phenotype can be reversed in patients by the drug isoproterenol, known for its β-adrenergic stimulatory effects, increasing calcium transport through L-type calcium channels, ryanodine receptors, and SERCA (via relief of phospholamban inhibition). Therefore, the function of calcium channels, including in the presence of post-translational modifications, should be investigated in BrS [2]. 

Currently included in BrS diagnostic panels are the calcium-related genes *CACNA1C*, *CACNA2D1*, *CACNB2*, *TRPM4*, and *PKP2* [98]. The *CACNA1C*, *CACNA2D1*, and *CACNB2* genes encode for the voltage-dependent L-type calcium channel subunits alpha 1C, alpha 2/delta subunit 1, and beta 2, respectively. Variants in *CACNA1C* have been reported to account for approximately 6.6% of BrS cases, while *CACNB2b* variants account for about 4.8%, and *CACNA2D1* variants are rare [102,103]. 

Variants in the *TRPM4* gene account for about 6% of BrS cases [58] and encode the Transient Receptor Potential Cation Channel Subfamily M Member 4, a calcium-activated nonselective ion channel that transports monovalent cations, such as Na^+^ and K^+^, across the membrane, increasing its activity in response to an increased intracellular calcium concentration, but without significant permeation of Ca^2+^ itself [104]. This might be one reason why it should be regarded carefully before being considered a real BrS gene. In spite of that, it is included in BrS diagnostic panels [98], described as potentially causative of BrS type 1 in an autosomal recessive, rather than an autosomal dominant, manner [13]. The *TRPM4* gene encodes for the protein Transient receptor potential cation channel subfamily M member 4, also called Calcium-Activated Non-Selective Cation Channel 1, which is a calcium-activated, but calcium-impermeable, nonselective cation channel [105]. Pathological variants in this gene have been implicated in complete heart block, ventricular tachycardia, and BrS [106], and this gene also plays a role in hypertrophy [107]. *TRPM4* gene protein products can influence the inotropic effect of β-adrenergic stimulation [105]. The half-life of its protein products have been reportedly altered in patients with complete heart block or ventricular arrhythmias [106]. Despite this, the role of the *TRPM4* gene in BrS must be interpreted carefully, as systematic, evidence-based evaluations for its causative effects are still lacking [17]. 

The *PKP2* gene, encoding plakophilin-2, found in desmosomes within the intercalated discs, links cadherins to intermediate filaments in the cytoskeleton, and has been specifically reviewed previously [108]. This gene was described in one report as being associated with approximately 2.5% of BrS cases in which patients did not harbor mutations in the BrS-related genes *SCN5A*, *CACNA1C*, *GPD1L*, or *MOG1* [59].

In a *PKP2* knockout mouse model, *RYR2*, *ANK2*, *CACNA1C*, and *TRDN* expression were reduced, as well as protein levels of calsequestrin-2, leading to disruption of intracellular calcium homeostasis and isoproterenol-induced arrhythmias because of loss of cell-cell communication [109]. However, the role of variants in this gene is disputed, as family segregation studies are lacking [17,110]. 

## 5. Potassium Channel Mutations

Variants in genes encoding potassium channels have been associated with BrS in a few cases, but their incidence needs to be assessed because not all cardiogenetics centers include these genes in the NGS panel. 

The *ABCC9* gene encodes for ATP Binding Cassette Subfamily C Member 9 transport proteins that carry various molecules across cell membranes, and variants in this gene have been reported to be associated with 4–5% of BrS cases [103]. 

The *HCN4* gene encodes the Hyperpolarization Activated Cyclic Nucleotide Gated Potassium Channel 4, important for pacing the heart rate. Variants in the *HCN4* gene are rarely associated with BrS patients [103]. The *KCND2* and *KCND3* genes encode the proteins Potassium Voltage-Gated Channel Subfamily D Member 2 and 3, respectively, and they play a role in heart rate regulation. Variants in these genes are rarely associated with BrS patients [103]. 

The genes *KCNE3* and *KCNE5* encode the proteins Potassium Voltage-Gated Channel Subfamily E Regulatory Subunit 3 and 5, respectively. Variants in either are rarely associated with BrS [103,111]. Despite this, they are still included in BrS diagnostic panels in many cardiogenetics centers [98]. 

The *KCNH2* gene, encoding the protein Potassium Voltage-Gated Channel Subfamily H Member 2, has been described in association with approximately 1–2% of BrS cases [103], and is included in diagnostic panels [98]. Finally, the *KCNJ8* gene, encoding the protein Potassium Voltage-Gated Channel Subfamily J Member 8, is rarely associated with BrS [103], although included in diagnostic panels [98]. Thus, although potassium channel variants are rarely found in association with BrS, and family segregation and functional studies are lacking [17], these genes are still routinely screened for variants in BrS patients. 

## 6. Sarcomeropathies

While BrS has been classically regarded as a channelopathy caused by variants in genes encoding for channel proteins on the cellular membrane, recent studies have identified non-channel genes associated with the syndrome. Also, although BrS was for a long time considered a “purely electrical disease without structural abnormalities”, recent studies have now reported epicardial surface and interstitial fibrosis, reduced gap junction expression, and increased collagen, as well as reduced contractility and RV structural abnormalities consistent with ARVC involving predominantly the RV anterior wall [2,35,112]. Along these lines, a study by Mango et al. described the finding of a mutation in the *TPM1* gene, encoding the sarcomeric α-tropomyosin, as causative of an overlap syndrome resulting in both hypertrophic cardiomyopathy and BrS phenotypes [34]. While further studies are necessary to confirm the role of *TPM1* variants in BrS, this study could imply that BrS may not always originate as variations in the genes encoding for channel proteins, but may also result from sarcomeric gene variants that are important in the regulation of calcium homeostasis [82]. In fact, it has been extensively studied that alterations in sarcomeric proteins can lead to arrhythmias and sudden death [113,114,115]. In one recent report, *MYBPC3* was associated for the first time with BrS [32], while many other sarcomeric variants have been associated with sudden death [115]. Thus, while BrS is classically considered a channelopathy, recent studies have demonstrated an overlap with cardiomyopathy characteristics, including structural abnormalities and findings in alterations in genes that encode for sarcomeric proteins.

## 7. Other Genes

The gene *SEMA3A* is involved in neuronal development and function, and it is currently included in some BrS diagnostic panels [98]. A PubMed search for “SEMA3A AND Brugada syndrome” on October 7, 2019 resulted in only three results: a meta-analysis in which only two variants in *SEMA3A* were identified out of 128 publications reporting 43 genes potentially associated with BrS [116], a review [98], and functional studies on cardiomyocytes derived from human-induced pluripotent stem cells [67]. In fact, a recent study contested the validity of the *SEMA3A* gene being included in BrS diagnostic panels [17]. Similarly, while the gene *FGF12* is currently included in diagnostic panels [98], a PubMed search for “FGF12 AND Brugada syndrome” on October 7, 2019 resulted in only two results: a review [98] and a study in which a variant in *FGF12* was found in a single patient [117]. Finally, while the gene *SLMAP* is currently included in diagnostic panels [98], a PubMed search for “SLMAP AND Brugada syndrome” on October 7, 2019 resulted in only four results: a review [98], functional studies in transgenic mice [68], a meta-analysis that described finding two variants among 128 publications reporting 42 genes potentially associated with BrS [116], and the study to which the meta-analysis seems to be referring [118]. The *SLMAP* gene has been recently disputed as causative of BrS, lacking systematic, evidence-based evaluations as to the association between this gene and BrS, while in this same study, the genes *SEMA3A* and *FGF12* did not even meet the criteria to be evaluated to be determined whether they could be potentially causative of BrS [17]. 

## 8. Discussion

Although the majority of BrS cases remain undisputedly molecularly unconfirmed, BrS is still considered by many to be a Mendelian disease. Clinical assessments support that BrS is an inheritable syndrome, but genotype-phenotype data to determine which specific variants are responsible for the disease is generally lacking. Questions remain as to which genes are involved and what is the real contribution of every variant found by NGS. Although many studies have focused on sodium channel mutations, alterations in genes that encode for sodium channels are not found in about two-thirds of cases, a fact which highlights the need to expand the research beyond the sodium channel gene *SCN5A*. It has to be mentioned also that other mechanisms can be involved in BrS pathogenesis that could indirectly affect the sodium channel function, which do not originate as genetic mutations in the sodium channel gene itself. 

BrS has a complex pathogenesis based on a dysfunction of ion channels: the fast upstroke in phase 0 of the cardiac action potential depends on the Na_V_1.5 protein that works as a channel for sodium cations (Na^2+^) [103]. Variants in the *SCN5A* gene believed to cause BrS are thought to work by resulting in a slowing of conduction in the heart. Among these variants, some have been studied functionally, revealing that a loss of function in Na_V_1.5 can occur through different mechanisms, including decreased expression of the sodium channel protein (Na_V_1.5) in the sarcolemma [119], production of a non-functional channel [33], or alteration in gating properties, such as delayed activation, earlier inactivation, faster inactivation, enhanced slow inactivation, and delayed recovery of Na_V_1.5 after inactivation [120,121].

There is emerging evidence about the role of CNVs involving the *SCN5A* gene as a cause of BrS. For instance, Mademont-Soler and colleagues described a duplication (from exon 15 to 28 of *SCN5A*) demonstrated with MLPA (multiplex ligation-dependent probe amplification) that was not detected in six other first-grade relatives, all negative to Flecainide challenge test [122]. Later, Sonoda and coworkers described, using the same technique, four BrS patients harboring CNVs in the *SCN5A* gene (3 deletion and 1 duplication) [123]. These are examples of a possible new mechanism for BrS, and further studies are needed to clarify the clinical significance of CNVs in the *SCN5A* gene in BrS patients.

While some BrS cases can be explained by either *SCN5A* variants or CNVs, not all BrS cases can be justified by a functional impairment in the sodium channel Na_V_1.5. Other sodium, calcium, or potassium channels can be involved in the disease’s pathogenesis. In particular, the role of calcium channels is an emerging field in which many groups are still studying all over the world [124,125].

One of the most interesting results recently reached by Abdelsayed and colleagues is that *SCN5A* heterozygous mutated embryonic kidney cells show differences in calcium sensitivity [126]. Thus, it could be possible that elevated cytosolic calcium concentrations (common, for example, during physical exercise) exacerbate the BrS phenotype when the syndrome is caused by a heterozygous *SCN5A* mutation. Another important study reported a complex BrS inheritance in a family harboring *SCN5A* and *CACNA1C* mutations [127].

While it seems relevant that the sodium channel plays a pivotal role in the syndrome, it remains unclear how genetic and non-genetic factors can influence the function of the sodium channel, or what other channel proteins or non-channel proteins may be involved. Whole genome studies may be useful to identify new candidate genes, which could then be further assessed using family segregation and functional studies. Complications surrounding whole genome studies include an extraordinary number of incidental findings, and thus a large number of patients are needed to determine which variants may be related to BrS and which variants are most likely incidental findings. Thus, it is important for researchers to form collaborations with other institutions to increase the study patient population, as well as to combine data from individuals within the same family for family segregation analysis when family members have been followed at different institutions. Obtaining family segregation data may include asking patients to put healthcare providers in contact with family members, asking the family to send medical records, and contacting the institutions of the family members for collaboration and publication purposes. 

Mouse models to assess the effects of the *SCN5A* gene in cardiac arrhythmias have been previously reviewed [128]. Briefly, in a knockout mouse with targeted disruption of the *SCN5A* gene, homozygous knockout of *SCN5A* is embryonically lethal, while heterozygous (*SCN5A*+/−) mice exhibited normal survival, decreased atrial, atrioventricular, and ventricular conduction, and age-related deterioration of ventricular conduction due to fibrotic remodeling and redistribution of connexin 43 expression. In the same model, ventricular tachycardia occurred earlier in the right ventricular outflow tract, suggesting that arrhythmias originate in this area. Additionally, *SCN5A* genetic defects increase susceptibility to atrial fibrillation [128]. Langendorff-perfused *SCN5A*+/− hearts exhibited a greater arrhythmic tendency in the RVOT, attributed to a combination of reduced Na_V_1.5 expression and increased fibrosis in the RVOT [129]. Hearts from *SCN5A*+/− mice have been shown to have greater incidences of bundle branch block and greater prominence of late conducting components, which were particularly evident in male or older mice and coupled with fibrosis [130]. Another study showed age-related lengthening of the P-wave and PR- and QRS-interval duration in *SCN5A*+/− mice, which coincided with the presence of fibrosis in the ventricular myocardium of the older mice, along with heterogeneous expression of connexin 43, upregulation of hypertrophic markers, including beta-MHC and skeletal alpha-actin, and upregulation of genes encoding Atf3 and Egr1 transcription factors [131]. Yet another study reported that both *SCN5A* disruption and aging were associated with decreased heart rate variability, reduced sinoatrial node automaticity, slowed sinoatrial conduction, increased collagen and fibroblast levels, and upregulated transforming growth factor-β(1) (TGF-β(1)) and vimentin transcripts [132]. Although these studies on *SCN5A*+/− mice provide valuable insights as to the potential effects of this gene, these results must be interpreted with caution, given species differences that result in differing phenotypes between species as reviewed previously [133,134], and it is still unclear what the potential effects of individual variants in the *SCN5A* gene may be. As described above, even polymorphisms in the *SCN5A* gene may act as modifiers, influencing a complex pathway that ultimately results in a BrS phenotype through a series of a combination of factors. 

Although studies in genetically altered mice are generally useful, it is difficult, or even impossible, to apply these sorts of models to BrS, because the genetics of BrS are not understood, making it impossible to know which genetic variant should be used to create the genetic model. Even *SCN5A* models are of limited use, since *SCN5A* variants are not even found in the majority of patients, and also because some *SCN5A* variants are not even thought to be pathogenic. Therefore, it is necessary to understand both the genes and specific pathogenic variants involved in the pathogenesis of BrS in humans before it will be possible to create animal models using those genetic alterations. Currently, single-cell methods, such as the use of induced pluripotent stem cells, are commonly used to assess the effect of individual variants within particular genes found in patients [47]. However, findings from these studies should be interpreted with caution, since this model is limited by the functional immaturity of the cells [135,136]. 

When possible, whole genome testing should be performed on BrS patients, and molecular autopsies should be performed on victims of sudden death. It is important to evaluate also polymorphisms, non-coding variants, and mitochondrial genes, as these may influence the BrS phenotype. In one report, mutations in mitochondrial tRNA genes were identified in Iranian BrS patients [137]. The data from whole genome testing should be compiled to create massive databases to differentiate disease-causing differences from incidental findings. Thus, further studies should be done on patients to uncover genetic mutations and to compile family segregation analysis data, so that specific candidate variants in specific genes potentially causative of BrS can be identified. 

In conclusion, given the vast uncertainty of the role of most variants in most genes currently included in BrS diagnostic panels, caution must be taken when interpreting genetic test results. Furthermore, genetic testing currently cannot be exclusively relied upon to predict the clinical phenotype or to perform risk stratification of future arrhythmic events. BrS appears to not be a pure Mendelian disorder, but rather a common ECG pattern that results from a vast number of diverse molecular pathologies. Thus, genetic testing alone, at this point, is not sufficient to understand the complexity of this syndrome. Perhaps additional omics approaches, such as epigenomics, transcriptomics, proteomics, metabolomics, lipidomics, and glycomics, could shed light on this complex pathology. 

## 9. Conclusions

(1)The literature strongly suggests that the concept of a single causative gene with autosomal dominant inheritance may not be the case in BrS.(2)First of all, it is clear that BrS patients can harbor both mutations and common variants, all potentially clinically meaningful, especially in the presence of multiple variants within the same individual, which can then have a combined pathological effect.(3)Therefore, Brugada syndrome seems to be a multifactorial disease, which is affected by several loci, each of which are influenced by the environment.(4)The influence of environmental factors for BrS clinical pictures can be both random and/or related with specific genetic variants, for example involving alcohol metabolizing enzymes.(5)The classification of both BrS-associated mutations and common variants is not possible without a complete functional study with patch clamp and/or the voltage clamp technique.(6)This study is aimed to understand where, when, how, and why a certain group of variants can impact cardiac channel function in a way that is necessary and sufficient to cause the manifestation of the BrS ECG.(7)With these data it might be possible to shed a new light on the pathophysiology of the heart conduction system and on the real contribution of genetics for the BrS clinical picture.

## Figures and Tables

**Table 1 ijms-21-01687-t001:** Genes currently associated with Brugada syndrome.

	Pubmed Results as of October 16, 2019 for “Brugada Syndrome AND (Gene)”	Examples of Family Segregation Studies Performed for BrS?	Examples of Functional Studies Performed for BrS?
SCN5A	742	[16,19,28,42,43]	[44]
SCN10A	26	[45,46]	[45,46,47]
SCN1B	36	[48]: but suggests SCN1B is not a monogenic cause of BrS	[49,50](*SCN1Bb*)
SCN2B	11	N/A	[51]
SCN3B	17	N/A	[52]
RANGRF	4	N/A	[53,54,55]
GPD1L	11	N/A	[56]
CACNA1C	30	N/A	N/A
CACNA2D1	6	N/A	N/A
CACNB2	15	N/A	[57]
TRPM4	14	N/A	[58]
PKP2	18	[59]	[59]
ABCC9	6	N/A	[60]
HCN4	10	N/A	[61]
KCND2	4	N/A	N/A
KCND3	20	N/A	[62,63]
KCNE3	11	N/A	[64]
KCNE5	6	N/A	[65]
KCNJ8	9	N/A	[66]
TPM1	1	[34]	N/A
MYBPC3	4	N/A	N/A
SEMA3A	3	N/A	[67]
FGF12	2	N/A	N/A
SLMAP	4	N/A	[68]
HEY2	129	N/A	[18,69,70,71]
LRRC10	1	N/A	N/A

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
