# Peer review of "Brugada Syndrome: Oligogenic or Mendelian Disease?"

_ijms, 2020, doi:10.3390/ijms21051687_

Round 1

Reviewer 1 Report

In this review, the authors describe briefly the clinical field of Brugada syndrome disease and widely the genetic background. They separate in a very comfortable reading the different channels genes.

I have different comments to the authors:

It could be appropriate to cite that the clinical symptoms of some diseases as BrS are increasingly likely to manifest themselves with increasing age of risk individuals. This can be a reason of incomplete penetrance of certain variant in a family. For this reason the following clinical and the genetic actualization is necessary.

In the 142-143 lines where the authors say: On the other hand, when studying modifiers of the SCN5A gene, it is poorly understood how a mutation in a structural gene, encoding for instance a desmosomal protein, can affect channel function enough to cause BrS in the absence of a mutation or variant in the SCN5A gene. Thus, the genetics of BrS are difficult to elucidate. There is a manuscript where identified several genetic variants in structural genes in a cohort of BrS: Genetic Analysis of Arrhythmogenic Diseases in the Era of NGS: The Complexity of Clinical Decision-Making in Brugada Syndrome identified variants in genes as JUP, DSP, PKP2, and DSG2 previously associated with ARVC. Also, it has been demonstrated that this genes interact with the sodium channel: Desmosomal proteins, particularly plakophilin and desmoglein, interact with Nav1.5 19 This interaction could be a possible explanation of the observed variation in genes encoding desmosomal proteins in BrS in our study. Shy D, Gillet L, Abriel H. Cardiac sodium channel NaV1.5 distribution in myocytes via interacting proteins: the multiple pool model. Biochimica et biophysica acta. 2013; 1833(4):886–94. Epub 2012/11/06.doi: 0.1016/j.bbamcr.2012.10.026 PMID: 23123192.

The authors not cite the genomic copy number variations (CNVS) as possible responsible of BrS. There are increasingly recognized that they contribute to risk for human diseases and in general, CNV deletions show higher penetrance (more severe phenotype) than duplications and longer CNVs often have higher penetrance and/or more clinical features than smaller CNVs. There is a research manuscript where they talk about it: Mademont-Soler, I.; Pinsach-Abuin, M.L.; Riuro, H.; Mates, J.; Perez-Serra, A.; Coll, M.; Porres, J.M.; Del Olmo, B.; Iglesias, A.; Selga, E.; et al. Large Genomic Imbalances in Brugada Syndrome. PLoS ONE 2016, 11, e0163514.

About the genes involved in BrS. In the literature LRRC1 gene has also been associated with BrS:  Electrocardiographic Assessment and Genetic Analysis in Neonates: A Current Topic of Discussion. Georgia Sarquella-Brugada, Sergi Cesar, Maria Dolores Zambrano, Anna Fernandez-Falgueras, Victoria Fiol, Anna Iglesias, Francesc Torres, Oscar Garcia-Algar, Elena Arbelo, Josep Brugada, Ramon Brugada, Oscar Campuzano. Curr Cardiol Rev. 2019 Feb; 15(1): 30–37. Published online 2019 Feb. doi: 10.2174/1573403X14666180913114806

I suggest to include the HEY2 gene in the table of genes currently associated with BrS. Bezzina C.R., Barc J., Mizusawa Y., Remme C.A., Gourraud J.B., Simonet F., Verkerk A.O., Schwartz P.J., Crotti L., Dagradi F., et al. Common variants at SCN5A-SCN10A and HEY2 are associated with Brugada syndrome, a rare disease with high risk of sudden cardiac death. Nat. Genet. 2013;45:1044–1049. doi: 10.1038/ng.2712.

The KCND3 gene appears 2 times in the table 1.

About the nomenclature please, review the guidelines of the Human Genome Variation Society to name the genetic variants.

Author Response

Reviewer 1:

In this review, the authors describe briefly the clinical field of Brugada syndrome disease and widely the genetic background. They separate in a very comfortable reading the different channels genes.

I have different comments to the authors:

COMMENT: It could be appropriate to cite that the clinical symptoms of some diseases as BrS are increasingly likely to manifest themselves with increasing age of risk individuals. This can be a reason of incomplete penetrance of certain variant in a family. For this reason the following clinical and the genetic actualization is necessary.

RESPONSE: While we agree that the longer a particular person lives, the more likely they are to experience symptoms, what we actually think is happening is that the people with the less severe phenotypes are living the longest. BrS can actually affect people of every age, even while still in utero or shortly after birth (SIDS). In our experience, the people who live for decades completely asymptomatic seem to have the lesser severe phenotypes. With that said, of course the older someone gets, the more time there is for them to develop symptoms (for example, some people seem to never get sick, but if they live long enough, they will eventually get sick and get a fever, which worsens the BrS phenotype). Therefore, we would like to avoid confusing the reader, who might think that elderly patients might have a worse phenotype. Actually, it is the opposite: elderly people who lived asymptomatic for decades probably have a less severe form.

COMMENT: In the 142-143 lines where the authors say: On the other hand, when studying modifiers of the SCN5A gene, it is poorly understood how a mutation in a structural gene, encoding for instance a desmosomal protein, can affect channel function enough to cause BrS in the absence of a mutation or variant in the SCN5A gene. Thus, the genetics of BrS are difficult to elucidate. There is a manuscript where identified several genetic variants in structural genes in a cohort of BrS: Genetic Analysis of Arrhythmogenic Diseases in the Era of NGS: The Complexity of Clinical Decision-Making in Brugada Syndrome identified variants in genes as JUP, DSP, PKP2, and DSG2 previously associated with ARVC. Also, it has been demonstrated that this genes interact with the sodium channel: Desmosomal proteins, particularly plakophilin and desmoglein, interact with Nav1.5 19 This interaction could be a possible explanation of the observed variation in genes encoding desmosomal proteins in BrS in our study. Shy D, Gillet L, Abriel H. Cardiac sodium channel NaV1.5 distribution in myocytes via interacting proteins: the multiple pool model. Biochimica et biophysica acta. 2013; 1833(4):886–94. Epub 2012/11/06.doi: 0.1016/j.bbamcr.2012.10.026 PMID: 23123192.

RESPONSE: We have now included these references into our manuscript, writing, “On the other hand, when studying modifiers of the SCN5A gene, it is poorly understood how a mutation in a structural gene, encoding for instance a desmosomal protein, can affect channel function enough to cause BrS in the absence of a mutation or variant in the SCN5A gene. Variants in genes encoding for desmosomal proteins, such as JUP, DSP, PKP2, and DSG2, have been associated with ARVD/C and dilated cardiomyopathy (DCM), both of which are associated with SCD (PMID: 26230511). Desmosomal proteins, such as plakophilin-2 (encoded by PKP2) and desmoglein-2 (encoded by DSG2), have been described to interact with NaV1.5 (PMID: 23123192) and implicated as a possible cause of BrS and ARVD/C (PMID: 24548564). However, the role of these desmosomal proteins in BrS pathology is still under debate.”

COMMENT: The authors not cite the genomic copy number variations (CNVS) as possible responsible of BrS. There are increasingly recognized that they contribute to risk for human diseases and in general, CNV deletions show higher penetrance (more severe phenotype) than duplications and longer CNVs often have higher penetrance and/or more clinical features than smaller CNVs. There is a research manuscript where they talk about it: Mademont-Soler, I.; Pinsach-Abuin, M.L.; Riuro, H.; Mates, J.; Perez-Serra, A.; Coll, M.; Porres, J.M.; Del Olmo, B.; Iglesias, A.; Selga, E.; et al. Large Genomic Imbalances in Brugada Syndrome. PLoS ONE 2016, 11, e0163514.

RESPONSE: We now have added, “There is emerging evidence about the role of CNVs involving the SCN5A gene as a cause of BrS. For instance, Mademont-Soler and colleagues described a duplication (from exon 15 to 28 of SCN5A) demonstrated with MLPA (Multiplex Ligation - dependent Probe Amplification) that was not detected in six other first-grade relatives, all negative to Flecainide challenge test (PMID: 27684715). Later, Sonoda and coworkers described, using the same technique, four BrS patients harboring CNVs in the SCN5A gene (3 deletion and 1 duplication) (PMID: 29574140). These are examples of a possible new mechanism for BrS, and further studies are needed to clarify the clinical significance of CNVs in the SCN5A gene in BrS patients.”

COMMENT: About the genes involved in BrS. In the literature LRRC10 gene has also been associated with BrS:  Electrocardiographic Assessment and Genetic Analysis in Neonates: A Current Topic of Discussion. Georgia Sarquella-Brugada, Sergi Cesar, Maria Dolores Zambrano, Anna Fernandez-Falgueras, Victoria Fiol, Anna Iglesias, Francesc Torres, Oscar Garcia-Algar, Elena Arbelo, Josep Brugada, Ramon Brugada, Oscar Campuzano. Curr Cardiol Rev. 2019 Feb; 15(1): 30–37. Published online 2019 Feb. doi: 10.2174/1573403X14666180913114806 (PMID: 30210005)

RESPONSE: We have now included the LRRC10 gene and this reference in our manuscript. We write, “Some centers now consider the LRRC10 gene to be associated with BrS (PMID: 30210005).  This gene encodes for the Heart-Restricted Leucine-Rich Repeat Protein and has been described as a transcriptional target of Nkx2.5, which regulates the ion channel proteins encoded by the SCN5A, CACNA1C, and KCNH2 genes (PMID: 28032242).” We also now include this gene in table 1.

COMMENT: I suggest to include the HEY2 gene in the table of genes currently associated with BrS. Bezzina C.R., Barc J., Mizusawa Y., Remme C.A., Gourraud J.B., Simonet F., Verkerk A.O., Schwartz P.J., Crotti L., Dagradi F., et al. Common variants at SCN5A-SCN10A and HEY2 are associated with Brugada syndrome, a rare disease with high risk of sudden cardiac death. Nat. Genet. 2013;45:1044–1049. doi: 10.1038/ng.2712. (23872634)

RESPONSE: We now include the HEY2 gene in table 1.

COMMENT: The KCND3 gene appears 2 times in the table 1.

RESPONSE: We have corrected this.

COMMENT: About the nomenclature please, review the guidelines of the Human Genome Variation Society to name the genetic variants.

RESPONSE: We cited the variants presented in the review just the way they were cited by our source, adding additional identifying information when possible, according to Varsome and the guidelines of the Human Genome Variation Society.

Reviewer 2 Report

Monasky et al. have done an extensive review of the genetics of Brugada syndrome. This paper is well described and provides important information to the readers. I also agree that Brugada syndrome should no longer be described as a single disorder, but rather defined as a group of disorders clumped together only by their common alteration in the ECG. The authors have done a commendable job in reviewing recent progress in the mechanism of Brugada syndrome.

Author Response

Thank you so much for your excellent review.